# Can LLMs Really Learn to Translate a Low-Resource Language from One Grammar Book?

**Seth Aycock**[1,2]   **David Stap**[2]   **Di Wu**[2]   **Christof Monz**[2]   **Khalil Sima'an**[1]

[1]ILLC, University of Amsterdam [2]LTL, University of Amsterdam
s.aycock@uva.nl

## Abstract

Extremely low-resource (XLR) languages lack substantial corpora for training NLP models, motivating the use of all available resources such as dictionaries and grammar books. *Machine Translation from One Book* (Tanzer et al., 2024) suggests that prompting long-context LLMs with one grammar book enables English–Kalamang translation, an XLR language unseen by LLMs—a noteworthy case of linguistics helping an NLP task. We investigate the source of this translation ability, finding almost all improvements stem from the book's parallel examples rather than its grammatical explanations. We find similar results for Nepali and Guarani, seen low-resource languages, and we achieve performance comparable to an LLM with a grammar book by simply fine-tuning an encoder-decoder translation model. We then investigate *where* grammar books help by testing two linguistic tasks, grammaticality judgment and gloss prediction, and we explore *what kind* of grammatical knowledge helps by introducing a typological feature prompt that achieves leading results on these more relevant tasks. We thus emphasise the importance of task-appropriate data for XLR languages: parallel examples for translation, and grammatical data for linguistic tasks. As we find no evidence that long-context LLMs can make effective use of grammatical explanations for XLR translation, we conclude data collection for multilingual XLR tasks such as translation is best focused on parallel data over linguistic description.

## 1 Introduction

Most of the world's languages are extremely low-resource (XLR), severely lacking in suitable corpora for NLP tasks (Ranathunga & de Silva, 2022), such as parallel data for machine translation (MT). However, over 50% of languages have both a dictionary and a grammar (Nordhoff & Hammarström, 2011). While human-readable, grammar texts are difficult to incorporate into most NLP models due to their non-standard, unstructured format. Large language models (LLMs) can handle free-form textual instructions and provide a potential solution to this data mismatch. After pre-training on trillions of tokens (in mainly high-resource languages), LLMs can learn tasks from only a few in-context examples (Brown et al., 2020; Wei et al., 2022). Given this, interest in exploiting grammar texts in-context for NLP tasks is growing (Ramos et al., 2024; Tanzer et al., 2024; Zhang et al., 2024b).

*Machine Translation from One Book* (Tanzer et al., 2024) claims LLMs can learn to translate between Kalamang (ISO 639-3: kgv)—a newly-documented language unseen in LLM training data—and English (eng) via *in-context learning* with only a grammar book. We note that kgv has over 3,000 parallel sentences, a dictionary with over 3,000 definitions (Visser, 2020), a 500-page grammar book (Visser, 2022) consisting of grammatical explanations and over 1000 parallel glossed examples, and nearly 100 typological feature specifications (Skirgård et al., 2023a;b). This level of resources is comparable to or more than thousands of XLR languages have (Joshi et al., 2020; OLAC, 2024), thus we expect most of these are also minimally represented in LLMs' pretraining data. Given this, finding methods to effectively exploit the available kgv resources could have wide-reaching implications for XLR NLP. In this paper, we question the claimed utility of grammatical explanations for XLR MT with LLMs, then ask *where* and *what kind* of grammatical knowledge helps. We show that:

**Parallel examples are essential for translation**  We disentangle grammar books' parallel examples from grammatical explanations, finding explanations add no *significant* advantage over parallel data: adding $+0.7$ CHRF++ into `kgv`, and into `eng` scores fall $-0.3$ points adding explanations to parallel sentences; and quality drops up to 8 points with parallel data removed. Our findings generalise to Nepali (`npi`) and Guarani (`gug`), where the book's parallel sentences outperform the full book by up to 4 CHRF++. LLMs fail to effectively exploit grammatical explanations *for translation*.

**Fine-tuning matches long-context LLMs**  We fine-tune small translation models on the parallel data, achieving competitive results within 0.2 CHRF++ of the performance of `Gemini` with a grammar book into `kgv`, and beating `Llama-3.1-8B` settings with access to the same data by up to 20 points. Parallel examples (especially with glosses) are both more *token-efficient* and readily available than grammar books, and enable computationally cheaper methods than long-context LLMs.

**Typological prompting outperforms explanations and helps linguistic tasks**  We introduce a novel typological feature prompt, and for `kgv` and `npi` translation we find our method is more effective than explanations into `eng`, but not into XLR languages. On `kgv` grammaticality judgment, our typological prompt improves up to 3% over the book's 1.2k parallel sentences and 8% over the whole book. For gloss prediction, parallel sentences again beat the book by up to 5.3% morpheme accuracy, and adding typology achieves leading performance on this task. Therefore LLMs *can* exploit grammar for *relevant* linguistic tasks—if provided in a useful *form*—but not for translation.

Task-appropriate data is therefore essential. In the current paradigm, we recommend that data collection for XLR MT is thus better focused on parallel data over linguistic description, given the advantages in token efficiency, computational cost, and availability.

## 2 RELATED WORK

**Grammar for low-resource machine translation**  Translation of low and extremely-low resource languages (here meaning $<$100k and 10k parallel examples respectively) with LLMs is currently of significant interest (Cahyawijaya et al., 2024; Court & Elsner, 2024; Iyer et al., 2024). Methods include fine-tuning (Xu et al., 2024), dictionary prompting (Ghazvininejad et al., 2023), and retrieval-augmented few-shot prompting (Merx et al., 2024). Alongside advances in long-context LLMs, recent work has introduced grammar information in context for various tasks: Guo et al. (2024) test a textbook-style prompt with LLM-generated parses, seeing limited gains against parallel sentences; and Zhang et al. (2024a) add a singular syntactic rule to their prompt with small effects. Zhang et al. (2024b) chain morphological analysis, a dictionary and an LLM-summarised grammar book in-context, observing small gains from book passages over a dictionary-only setup. Others meanwhile use grammars with LLMs for data augmentation (Lucas et al., 2024) or as a hybrid rule-based translation system (Coleman et al., 2024).

*Machine Translation from One Book* (MTOB) (Tanzer et al., 2024) introduces a translation test set for the newly documented XLR language Kalamang (thus unseen by LLMs), plus a grammar book and additional parallel sentences. MTOB suggests long-context LLMs can exploit linguistic knowledge (a grammar book) for XLR translation, a potential step forward in leveraging underused resources for XLR languages. However, several issues mean MTOB leaves open questions over LLMs' ability to exploit linguistic information for XLR tasks. The test sets of 50 short, easy examples are potentially too small for making wider generalisations, and the human baseline is somewhat flawed as they may learn from examples at test time; relatedly, Gemini Team et al. (2024) ask the non-fluent human baseline to rate model outputs and their own `kgv` predictions, potentially biasing the evaluation. Furthermore, despite CHRF++ being the de-facto standard in XLR translation (Maillard et al., 2023; Costa-jussà et al., 2024; Edman et al., 2024), MTOB uses CHRF which unlike CHRF++ does not factor in word order. MTOB's results would benefit from further ablations, since the signal from parallel sentences and explanations is not disentangled, nor is a strong translation approach tested. Finally, we note that the `kgv` grammar book is not designed for language learning, but for describing theoretical linguistic phenomena—which MTOB's authors note limits LLMs to a basic competence. In this paper, we tackle these issues by combining the test sets, using automatic CHRF++ scores, disentangling the parallel/non-parallel signal, and testing two tasks better aligned for grammar books.

**Linguistics in NLP**  Incorporating linguistic information into NLP models is a long-standing goal with mixed results (Lakoff, 1978; Raskin, 1985; Uszkoreit, 2009; Opitz et al., 2024). Past work sees

gains from adding syntactic knowledge into translation models using constituency parses (Currey & Heafield, 2019), grammar supertags (Nădejde et al., 2017), or tree-structured models (Sartran et al., 2022). One useful form of linguistic information is *typology*, available for many languages in standardised feature databases (Dryer & Haspelmath, 2013; Skirgård et al., 2023a;b); features describe languages in terms of phenomena such as word order rules, verb tenses, and noun cases. Trained linguists condense fine-grained textual descriptions from grammar books into discrete, categorical, and cross-linguistically consistent feature specifications. Typological features have been incorporated into NLP models with some success in the form of embeddings (Malaviya et al., 2017; Östling & Tiedemann, 2017) but several studies find minimal positive effects on performance (Ponti et al., 2019; Üstün et al., 2022). To test whether LLMs follow this trend, we construct a novel prompt that uses readily available typological feature specifications for source and target languages, as an in-context and language-invariant method for bridging cross-lingual grammatical differences.

**Interlinear gloss prediction** Language documentation involves describing the underlying grammar of a language given its surface forms (Ginn et al., 2023). A standardised data format for this analysis is *Interlinear Glossed Text* (IGT), comprising a morphologically segmented *transcription* (where morphemes are the smallest units of meaning), an aligned *interlinear gloss* with subword-level lexical and grammatical information, and a sentence-level *translation* (Comrie et al., 2015; Mortensen et al., 2023); a Kalamang example is shown in Example 1 (Visser, 2022). We note that glossing is designed for trained linguists rather than language learners. Glosses have been widely applied in NLP tasks, including as a pivot for translation (Zhou et al., 2020), dependency parsing (Georgi et al., 2012), grammar generation (Bender et al., 2014), morphological analysis (Moeller et al., 2020; Shandilya & Palmer, 2023), and linguistic resource development (Beermann et al., 2020; Buchholz et al., 2024). Predicting IGT is therefore a well-motivated grammatical task, and segmented IGT is a valuable linguistic resource. IGT prediction is most relevant for XLR languages where it is impactful in assisting annotators for documentation and preservation (Ginn et al., 2023). Prior methods include supervised neural models (Zhao et al., 2020; Girrbach, 2023), or adapting multilingual language models (He et al., 2023; Ginn et al., 2024a;b). Since IGT is costly to generate, past work has scraped it from books (Nordhoff, 2020; Nordhoff & Krämer, 2022); we follow this method to extract Kalamang glosses from the grammar book. One of our contributions involves testing gloss prediction to determine whether LLMs can use grammatical knowledge for more relevant tasks.

(1)  bal  se   sor=at  na      ma  se  nan=i      koyet      **Transcription**
     dog  IAM  fish=OBJ consume 3SG IAM consume=PLNL finish    **Interlinear Gloss**
     'The dog ate the fish, after he ate.'                      **Translation**

## 3 METHODOLOGY

### 3.1 GRAMMAR BOOKS FOR TRANSLATION

Our methods are guided by open questions over the use of grammar books for XLR translation. First, we manually filter the grammar books into parallel examples and word pairs, and explanatory, descriptive text, to disentangle the signal from translations and grammatical explanations (see Appendix A for a `kgv` book extract). This novel ablation is necessary to understand which specific aspects of grammar books are useful for XLR MT. We ask whether LLMs really learn effectively from the grammar explanations, or if most translation supervision stems only from the book's parallel examples. We combine the directional test sets into a single 100 example test set to improve the generalisability of these results, and evaluate with CHRF++ (Popović, 2017) to take word order into account[1]. We also test `eng–npi` and `gug` translation, low-resource languages with an established evaluation set, FLORES (Costa-jussà et al., 2024) and likely a low data weight in LLMs; while not unseen, these experiments broaden our results to seen low-resource languages more generally.

### 3.2 NEURAL MACHINE TRANSLATION APPROACHES

To compare the LLM-based approach with a standard MT approach for learning to translate a language as yet unseen by the model, we run experiments fine-tuning NLLB-1.3B (Costa-jussà et al., 2024)

---

[1] We omit human evaluation (cf. Gemini Team et al., 2024) given the infeasibility of engaging proficient Kalamang speakers. See Appendix I for a small-scale qualitative analysis of several `kgv–eng` examples.

on the parallel data sourced from the grammar book. We expect similar results to be achieved with the same resources using a small, specialist encoder-decoder model, which would confirm that the useful translation signal stems from the parallel sentences contained within grammar books—which constitute less than 20% of the `kgv` grammar book's total tokens (see Table 1 for token counts).

### 3.3 TYPOLOGICAL FEATURE PROMPTING

In asking *what kind* of grammatical knowledge can aid LLMs in XLR tasks, we introduce a text-based method for incorporating typological information into prompts, differing from previous work on continuous typological embeddings (Oncevay et al., 2020). We extract categorical typological feature specifications from Grambank Skirgård et al. (2023b) for `kgv`, `npi`, `gug`, and `eng`, and use a rule-based template to construct a prompt containing features for each language and a short explanation. For an example of the prompt format, see Appendix D. Most languages with grammar books have some typological feature specification, since features are distilled by annotators from external resources. Our method isolates high-level grammatical tendencies of a language from the specific instantiations of those features (i.e. parallel examples). We hypothesise that our method, when combined with the grammar book's parallel sentences, will at least match the performance of the grammar book. We expect that providing explicit features such as word order rules removes some reasoning requirements for the LLM. Conversely, typological features will not have *relevant* parallel examples, so some reasoning and retrieval is still required, potentially tempering the advantages.

### 3.4 GRAMMATICALITY JUDGMENT

To test the LLM's ability to acquire knowledge and understanding of Kalamang grammar from the book, we introduce a discriminative grammar judgment experiment. We ask the model to choose the original Kalamang test sentence against a modified example, with three successively easier settings: swapping two adjacent words ($\text{SWAP}_{\text{adj}}$), two random words ($\text{SWAP}_{\text{ran}}$), and shuffling all words ($\text{SHUFFLE}$). We acknowledge that while we cannot guarantee all corruptions are ungrammatical (since no author speaks Kalamang), we assume the uncorrupted examples are linguistically unmarked sentences. For all settings we expect a 0-SHOT model to achieve approximately 50% accuracy, while for high-resource languages we would expect near 100% accuracy. We expect the grammar book to have a greater positive impact in this setting where grammatical knowledge is explicitly rewarded.

### 3.5 INTERLINEAR GLOSSED TEXT PREDICTION

To explore another more relevant task for exploiting grammar explanations, we test IGT prediction with the grammar book against few-shot and supervised baselines. This experiment tests whether LLMs can learn grammar from a book to the extent that we see a difference in performance on a grammar-focused task. IGT requires both lexical translation and grammar analysis, without any generation in the language at hand. This makes IGT prediction a more appropriate task to perform from a descriptive, non-didactic grammar text. We argue that IGT prediction accelerates XLR documentation more than translation, and is likely to have more direct impact for both first language (L1) speakers and linguists, not to mention the potential downstream uses, e.g. POS tagging and MT. IGT prediction is also a well defined task with strong baselines from a shared task (Ginn et al., 2023) and clear evaluation metrics, primarily morpheme accuracy (McMillan-Major, 2020; Zhao et al., 2020); our experiments build on this prior work. Finally, we argue grammar books are intuitively suited to IGT prediction more than translation, because their unique contribution is glossed text, rather than just parallel sentences. We use all available sentences with IGT from Dictionaria as our test set, and for our supervised baselines, we process the grammar book IGT examples into a training and development set. We expect the grammar book to provide marginal gains over raw parallel sentences because the grammar book explicitly explains the glossed examples therein.

## 4 EXPERIMENTAL SETUP

### 4.1 DATA

We use the preprocessed Kalamang (`kgv`) grammar book (Visser, 2022; Tanzer et al., 2024), with additional processing of irregularities (particularly for glossing) introduced in LaTeX conversion.

We similarly preprocess a grammar text in Nepali (`npi`) (Bal, 2004) and Paraguayan Guarani (`gug`) (Estigarribia, 2020). We prompt with the entire grammar, BOOK$_{all}$, in-context (where the subscript indicates the data subset). Following Nordhoff & Krämer (2022), we extract parallel glossed examples and bilingual word/phrase pairs from the book based on text formatting into a parallel subset, BOOK$_{para\ (p)}$. The remainder of the book contains grammatical explanations without parallel examples, labelled BOOK$_{non-para\ (\neg p)}$. Subset statistics are shown in Table 1. We preprocess `kgv-eng` parallel examples from the grammar book into an unsegmented parallel data format, giving PARA$_{book}$ (used for 5*-SHOT examples and in full as a prompt) and PARA$_{book}^{IGT}$ which includes glosses (1239 examples) – for excerpts of prompt types, see Appendix E. We additionally test prompts with PARA$_{train}$ (400 examples) and WORDLIST (W) (3813 examples). Additionally, we sample 500 examples from Dictionaria[2] (Visser, 2020) as the development set for fine-tuning. In total there are 3.3k eng$\rightleftharpoons$kgv parallel examples[3]; we focus on the 1.2k in PARA$_{book}$ for fair comparison with BOOK settings. For testing, we use our combined 100 example test set for `kgv`, and FLORES devtest for `npi` and `gug` (1012 examples) (Guzmán et al., 2019), with few-shot examples from FLORES dev. For IGT prediction, we preprocess 1221 examples from the grammar book with glosses for training (5623 words) and development (612 words) sets (split 90:10% by sentences). Following Ginn et al. (2023), we introduce a test set of 97 glossed examples (447 words) from a different source, Dictionaria, which were manually inspected for correct alignment.

Table 1: Dataset statistics for grammar book subsets, in lines and space-separated tokens.

| Language | Split | Lines | Tokens |
|---|---|---|---|
| `kgv` | BOOK$_{para}$ | 4489 | 17858 |
| | BOOK$_{non-para}$ | 2282 | 81268 |
| `npi` | BOOK$_{para}$ | 759 | 5333 |
| | BOOK$_{non-para}$ | 2896 | 23233 |
| `gug` | BOOK$_{para}$ | 5718 | 49122 |
| | BOOK$_{non-para}$ | 3295 | 57338 |

## 4.2 MODELS

In our experiments we use the API-only Gemini-1.5-Flash-001 (henceforth `Gemini`) (Gemini Team et al., 2024). We justify this choice due to `Gemini`'s context window of 1M tokens, significantly larger than other models, which can handle the entire grammar book, and use `Flash` over `Pro` due to prohibitive cost differences. We also use the smaller, open-weight Llama-3.1-8B base and instruction-tuned models (Dubey et al., 2024), with a context of 128k tokens. This is insufficient for `kgv` and `gug` BOOK$_{all}$, but fits BOOK$_{para}$ and BOOK$_{non-para}$, plus the `npi` BOOK$_{all}$. We test Llama-Instruct (`Llama-I`), and fine-tune Llama base with LoRA (Hu et al., 2021) on PARA$_{book}$ (`Llama-ft`) with prompt masking for 5 epochs with a constant learning rate of 1e-4, batch size 4, and LoRA $\alpha = 16$, $r = 16$, targeting all linear projections. For our NMT baseline, we fine-tune NLLB-1.3B-Distilled (`NLLB`) (Costa-jussà et al., 2024) on PARA$_{book}$. For `kgv` grammaticality judgment and IGT prediction, we use the same `Gemini` model as above.

## 4.3 EVALUATION

We evaluate translation automatically with CHRF++ (Popović, 2017). We favour CHRF++ over CHRF, used in Tanzer et al. (2024), since it takes into account word order as well as character $n$-gram overlap. We report scores for trimmed responses after the first newline character to distinguish translation quality from overgeneration and chat explanations (Aycock & Bawden, 2024) and use a forceful prompt (detailed in Appendix E) to ensure the translation is produced on the first line.

---

[2] `https://dictionaria.clld.org/contributions/kalamang`
[3] Data (including grammar book splits) and code are made available at this link.

Table 2: Translation results for `eng⇌kgv` with `Gemini`, Llama-Instruct (`L-I`) and fine-tuned (`L-ft`), and prompt tokens counted with NLTK's tokenizer (Bird et al., 2009). Highest $\text{BOOK}_{\text{para}}$ scores are underlined, highest overall are **bolded**. Grey rows indicate settings with data other than the book's parallel data; – indicates tests ruled out by context length. *W4W tests are not run with `Gemini` but are included for comparison. The subset of the book's parallel sentences almost matches or outperforms the whole grammar book, while its grammatical explanations perform poorly.

| Setting↓ | CHRF++ | | | | | | Tokens |
|---|---|---|---|---|---|---|---|
| | eng–kgv | | | kgv–eng | | | |
| Model→ | Gemini | L-I | L-ft | Gemini | L-I | L-ft | |
| **BASELINES** | | | | | | | |
| 0-SHOT | 11.0 | 2.7 | 18.5 | 12.7 | 12.5 | 23.0 | 0 |
| W4W | 18.9* | – | – | 18.2* | – | – | 0 |
| **PARALLEL DATA** | | | | | | | |
| WORDLIST (W) | 29.1 | 13.6 | 19.5 | 27.9 | 20.8 | 26.8 | 9.0k |
| 5*-SHOT $\text{PARA}_{\text{book}}$ | 38.9 | 15.0 | 24.6 | 33.4 | 21.1 | 23.0 | 0.8k |
| $\text{PARA}_{\text{book}}$ | 26.6 | 7.3 | 13.0 | 33.1 | 22.9 | 26.9 | 15.6k |
|   + W | 34.7 | 6.8 | 14.4 | 34.7 | 27.5 | 30.5 | 24.6k |
|     + $\text{PARA}_{\text{train}}$ | 40.7 | 13.8 | 17.9 | **46.6** | **31.3** | **37.6** | 29.4k |
| $\text{PARA}_{\text{book}}^{\text{IGT}}$ | 33.7 | **20.3** | **28.8** | 32.8 | 24.7 | 33.1 | 22.7k |
| **GRAMMAR BOOK SUBSETS** | | | | | | | |
| $\text{BOOK}_{\text{all}}$ | 34.4 | – | – | 34.4 | – | – | 99.6k |
|   + W | 38.3 | – | – | 39.6 | – | – | 108.6k |
|     + $\text{PARA}_{\text{train}}$ | **43.7** | – | – | 46.1 | – | – | 113.4k |
| $\text{BOOK}_{\text{para}}$ $(p)$ | 30.8 | 9.7 | 19.0 | 34.7 | 22.1 | 28.8 | 18.3k |
| $\text{BOOK}_{\text{non-para}}$ $(\neg p)$ | 22.6 | 3.3 | 10.0 | 27.5 | 14.3 | 16.7 | 81.3k |
| **TYPOLOGY** | | | | | | | |
| TYP 0-SHOT | 10.8 | 3.4 | 13.6 | 13.9 | 14.3 | 17.6 | 68.4k |
|   + $\text{BOOK}_{\text{para}}$ | 31.4 | – | – | 35.2 | – | – | 86.7k |
|   + $\text{PARA}_{\text{book}}^{\text{IGT}}$ | 32.9 | – | – | 33.0 | – | – | 84.0k |
|   + W + $\text{PARA}_{\text{book+train}}$ | 40.6 | – | – | 44.9 | – | – | 100.6k |

## 4.4 BASELINES

For translation experiments, we test several baselines: 0-SHOT translation with a standard translation prompt; word-for-word translation with fuzzy dictionary lookup (W4W); 5 retrieved examples *per word* (5*-SHOT) based on longest common subsequences following Tanzer et al. (2024); prompting with the full WORDLIST (W), parallel examples, $\text{PARA}_{\text{book}}$, parallel examples with glosses, $\text{PARA}_{\text{book}}^{\text{IGT}}$, and processed training set examples, $\text{PARA}_{\text{train}}$. For IGT prediction, we use a baseline frequency-based classifier (TOP-CLASS), a fine-tuned RoBERTa token classifier (Ginn et al., 2023) (SMP-BASE); a hard-attention glossing model (TÜCL-MORPH) (Girrbach, 2023); and BYT5-FT and GLOSSLM-FT models (Ginn et al., 2024b) fine-tuned on our `kgv` IGT training and development sets. We provide segmented input, and English translations to models which accept them.

## 4.5 EXPERIMENTS

Our central research question investigates the contributions of grammatical explanations and parallel data to translation performance. We therefore prompt models with $\text{BOOK}_{\text{all}}$ and its filtered subsets. We test our typological feature prompt, TYP, to replace $\text{BOOK}_{\text{non-para}}$. For `npi` and `gug`, we repeat the book settings as above. We fine-tune translation models with the $\text{PARA}_{\text{book}}$ parallel data for comparison with $\text{BOOK}_{\text{para}}$ settings. For grammaticality judgment and IGT prediction tasks, we similarly test `Gemini` with the `kgv` BOOK and TYP prompts.

Table 3: Translation results for `eng⇌npi` and `eng⇌gug` with `Gemini` and `Llama-I`. Best BOOK$_{\text{para}}$ (white rows) scores are underlined, best overall are **bolded**; – indicates tests ruled out by context length. While BOOK$_{\text{all}}$ and BOOK$_{\text{non-para}}$ decrease performance from 0-SHOT, BOOK$_{\text{para}}$ has a neutral or positive effect into and from `npi` respectively, with a similar trend seen for `gug`.

| Setting↓ | CHRF++ | | | | | | | |
|---|---|---|---|---|---|---|---|---|
| | eng–npi | | npi–eng | | eng–gug | | gug–eng | |
| Model→ | Gemini | L–I | Gemini | L–I | Gemini | L–I | Gemini | L–I |
| 0-SHOT | 42.5 | 28.6 | **65.2** | 51.1 | 26.6 | 6.1 | 41.3 | **23.6** |
| 5*-SHOT | **43.2** | **37.6** | 64.9 | **57.3** | **29.2** | **13.7** | **43.1** | 23.4 |
| BOOK$_{\text{all}}$ | 42.6 | 24.3 | 64.4 | 48.9 | 22.2 | – | 38.7 | – |
| BOOK$_{\text{para } (p)}$ | 42.5 | 28.6 | 64.9 | 52.6 | 25.8 | 6.7 | 41.8 | 11.8 |
| BOOK$_{\text{non-para } (\neg p)}$ | 41.8 | 24.5 | 64.5 | 48.4 | 19.3 | 5.6 | 34.5 | 10.1 |
| TYP 0-SHOT | 42.4 | 23.2 | 64.6 | 49.5 | 21.1 | 4.3 | 33.9 | 23.4 |
| TYP + BOOK$_{\text{para}}$ | 41.8 | 22.0 | 64.9 | 49.1 | 21.9 | – | 34.5 | – |

## 5  RESULTS & ANALYSIS

**Grammar versus parallel sentences for translation**   We disentangle the signal from grammar books' explanations and parallel sentences for translation. Our `kgv` results in Table 2 show that most or all performance improvements stem from the book's parallel sentences, with quality plummeting when parallel data is removed. With `Gemini` into `eng`, BOOK$_p$ marginally outperforms BOOK$_{\text{all}}$, and beats BOOK$_{\neg p}$ by 7 CHRF++, while into `kgv`, BOOK$_p$ outperforms BOOK$_{\neg p}$ by over 8 points, and BOOK$_{\text{all}}$ performs 3 points better than BOOK$_p$. However, we show statistically in Section 5.1 that this small improvement is modelled directly by an increase in test set vocabulary coverage, rather than from the grammatical explanations. Additionally, this gap closes with the PARA$_{\text{book}}^{\text{IGT}}$ prompt, which preprocesses and structures the parallel data in BOOK$_p$ into `kgv`–gloss–`eng` triples. PARA$_{\text{book}}^{\text{IGT}}$ performs particularly well for `Llama-I`, with over 10 points improvement over BOOK$_p$ into `kgv`. Due to context restricting `kgv` BOOK$_{\text{all}}$ tests, conclusions with `Llama-I` are limited, but we find again that BOOK$_{\neg p}$ performance lags far behind BOOK$_p$. We note that baselines including 0-SHOT show `kgv` translation is non-trivial. We also find that additional parallel data further improves translation quality, and note 5*-SHOT is generally competitive despite its short average prompt, achieving the best BOOK$_p$ score into `kgv` with `Gemini`. Thus for `kgv` translation, both LLMs on test mainly learn from the book's parallel sentences, failing to exploit the grammatical explanations.

We observe a similarly strong trend for `npi` and `gug`, seen low-resource languages with high-quality FLORES test sets, in Table 3. BOOK$_p$ settings largely match or outperform BOOK$_{\text{all}}$ for both models and languages (except `Llama-I` in `gug`–`eng` where the model often fails to output translations on the first line for BOOK settings). Few settings beat 0-SHOT and differences between `Gemini` settings (especially `npi`) are smaller than for `kgv`; perhaps the model's prior competence (and a shorter `npi` grammar book) mean there is less to be gained. However, analysing BOOK settings in isolation shows that both BOOK$_{\text{all}}$ and BOOK$_{\neg p}$ have a detrimental effect of up to 7 points below 0-SHOT, while BOOK$_p$ has a neutral or small positive impact in both `npi` and `gug`. Finally 5*-SHOT is again effective, especially for `Llama-I` into `npi` and `gug`, likely due to the greater vocabulary coverage of the example set. These results generalise our findings for `kgv` to seen low-resource languages: we find no evidence that LLMs can effectively exploit grammatical explanations *for translation*.

**Fine-tuning versus in-context learning**   We test a standard MT approach for adding a new language by fine-tuning `NLLB`, a small MT model, on the book's parallel data, shown in Table 4. `NLLB` achieves competitive or improved performance compared to prompting `Gemini` with the same preprocessed parallel data, PARA$_{\text{book}}$. We also test backtranslation (BT), a standard method to boost performance in MT (Sennrich et al., 2016). A single BT iteration with PARA$_{\text{train}}$ has a negative impact into `kgv`, likely due to the poor quality of the initial model introducing excessive noise. However we see a boost of 3 CHRF++ into `eng`, we expect because of the strong English language modelling of `NLLB`. Further, adding a small 400 example parallel training set sees large gains of 4-8 points. These results suggest the MTOB benchmark can be adequately addressed as a standard XLR MT problem with simple data preprocessing, a small pre-trained model, and fine-tuning on a single GPU for 1 hour.

Table 4: Translation results for eng⇌kgv with NLLB, an MT model, fine-tuned on PARA_book data; equivalent in-context learning results with Gemini are shown for comparison. Fine-tuned NLLB achieves competitive results with an LLM given the same parallel data, especially into kgv.

| Setting↓ | CHRF++ | | | |
| | eng–kgv | | kgv–eng | |
| Model→ | Gemini | NLLB | Gemini | NLLB |
|---|---|---|---|---|
| PARA_book | 26.6 | 34.2 | 33.1 | 28.6 |
| + PARA_train | 33.4 | 38.7 | 38.5 | 36.9 |
| + BT PARA_train | – | 32.0 | – | 31.6 |

We also fine-tune Llama base on PARA_book to give Llama-ft, with results in Table 2. We find all Llama-ft settings beat equivalent Llama-I tests with BOOK_all data, except for $\text{PARA}_{\text{book}}^{\text{IGT}}$ settings with glosses which marginally outperform Llama-ft 0-SHOT results. Prompting Llama-ft with parallel data in-context further improves performance over 0-SHOT by up to 10 points. We additionally fine-tune Gemini on PARA_book, with results in Appendix G, finding Gemini-ft underperforms NLLB and Gemini with the same data in-context by 6-12 CHRF++; we expect this is because it is already extensively instruction-tuned. Thus fine-tuning—particularly of small MT models—is a cheap method for achieving competitive results with prompting instruction-tuned long-context LLMs, given the same parallel data.

**Typological prompting for linguistic tasks** Given the limited contribution of grammatical explanations to translation performance, we introduce a novel prompting method summarising languages' typological features. This prompt is intended to replace BOOK_¬p, thus we are primarily focused on results when combined with BOOK_p data. Our results for eng–kgv translation in Table 2 show expectedly poor 0-SHOT performance due to the lack of any Kalamang text. Into kgv, our prompt beats BOOK_p but not BOOK_all; however into eng, our prompt with BOOK_p achieves the best translation results for settings with book parallel data. For npi in Table 3, TYP + BOOK_p is less effective than BOOK_all into npi, and marginally outperforms it into eng up to 0.5 CHRF++, though BOOK_p alone performs best; similarly in gug tests, BOOK_p outperforms TYP + BOOK_p, which beats or matches BOOK_¬p. The performance of typological prompting for translation is therefore inconsistent, supporting the above finding that LLMs fail to effectively exploit grammatical information *for MT*.

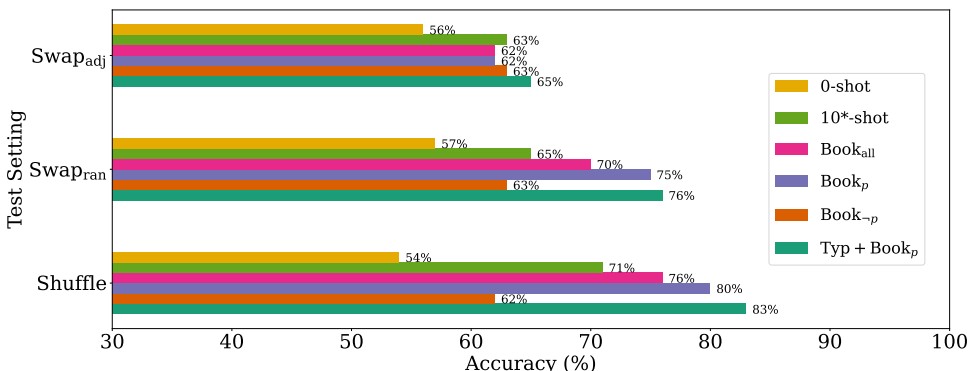

Figure 1: Grammaticality judgment accuracy in kgv; for reference in eng tests, Gemini scores 100%, 99%, and 100% respectively. Our prompt TYP + BOOK_p performs best overall suggesting grammar can help LLMs for linguistic tasks.

To determine whether grammar is not useful for MT or LLMs cannot exploit grammatical explanations more broadly, we test two more relevant tasks: grammaticality judgment and IGT prediction. In Figure 1, grammaticality judgment results in kgv with Gemini show all settings perform similarly poorly on SWAP_adj, though improving on 0-SHOT by around 7%. Generally, 10*-SHOT is worse than prompts with BOOK_p, likely because diverse sentences may help here more than overlapping

Table 5: IGT prediction results in `kgv` for supervised baselines and `Gemini` settings. Our TYP + BOOK$_{\text{para}}$ prompt achieves the highest morpheme accuracy and high scores on other metrics, while BOOK$_{\text{all}}$ performs poorly overall.

| Model | | Morph Acc. | Word Acc. | Stem F1 | Gram F1 | CHRF++ |
|---|---|---|---|---|---|---|
| TOP-CLASS | (Ginn et al., 2023) | 44.0 | 39.7 | 40.6 | 57.8 | 34.5 |
| SMP-BASE | (Ginn & Palmer, 2023) | 45.2 | 41.7 | 39.7 | **58.9** | 34.3 |
| TüCL-MORPH | (Girrbach, 2023) | 43.6 | 38.8 | 40.0 | 50.7 | 35.4 |
| BYT5-FT | (Xue et al., 2022) | 40.8 | **48.6** | 40.9 | 45.4 | 49.0 |
| GLOSSLM-FT | (Ginn et al., 2024b) | 43.8 | 47.7 | 41.5 | 50.4 | **49.1** |
| 10*-SHOT | | 43.9 | 43.7 | **44.3** | 45.2 | 46.4 |
| BOOK$_{\text{all}}$ | | 40.1 | 31.5 | 38.7 | 43.4 | 40.5 |
| BOOK$_{\text{para}}$ ($p$) | | 45.4 | 42.1 | 44.0 | 49.0 | 45.0 |
| BOOK$_{\text{non-para}}$ ($\neg p$) | | 21.0 | 8.8 | 23.9 | 15.6 | 26.0 |
| TYP + BOOK$_{\text{para}}$ | | **46.1** | 40.9 | 44.2 | 50.5 | 44.8 |

vocabulary, which helps more for MT. For BOOK settings we observe that BOOK$_p$ matches or outperforms BOOK$_{\text{all}}$ across all three tests by up to 5%, and consistently beats BOOK$_{\neg p}$, by up to 18% in SHUFFLE tests. So far, the LLM still fails to exploit grammatical explanations effectively and learns mainly from parallel examples. However, our TYP + BOOK$_p$ setting performs best over the three tests by up to 3% over BOOK$_p$. These positive results suggest that LLMs *can* learn from grammar, given the right kind of grammatical knowledge and a relevant task.

For `kgv` IGT prediction, we compare `Gemini` settings with supervised baselines in Table 5. The leading performer in morpheme accuracy, the key IGT metric, is again our typological prompt TYP + BOOK$_p$, scoring 6% above BOOK$_{\text{all}}$, 0.5% over BOOK$_p$, and 25% over BOOK$_{\neg p}$. Additionally, our prompt beats all supervised systems by 1-5%, suggesting in-context learning with typological knowledge and parallel glossed examples is a strong method for XLR IGT prediction. Results for other metrics show slightly differing trends, with supervised models showing stronger word accuracies and Gram F1 scores (since most are closed-set classifiers). Generally though, BOOK$_{\neg p}$ shows extremely poor performance, while BOOK$_p$, 10*-SHOT, and TYP + BOOK$_p$ settings perform consistently well, often beating supervised baselines. We note that TYP + BOOK$_p$ scores show competent performance for both grammatical (on morpheme accuracy and Gram F1) and lexical aspects (via Stem F1) of IGT prediction, suggesting all-round competence on this task. These results reinforce our findings that while parallel sentences still provide most of the useful signal, LLMs can exploit grammatical—specifically typological—information for linguistic tasks.

## 5.1 ANALYSIS

**Type coverage and Token efficiency** We investigate whether any added performance from grammatical explanations is statistically significant or can instead be attributed to greater test set type coverage in the prompt. We distinguish between types, meaning unique words in a vocabulary, and tokens, i.e. individual occurrences of types in a text. We fit univariate least squares regression models to CHRF++ scores with test set *type* coverage as the independent variable, for both directions, shown in Figure 2. All settings fall within the 95% confidence interval of the regression lines, and the models are significant in both directions ($p < 0.005$, F-test)[4]; the Pearson correlations are also significant ($p < 0.005$). Thus maximising target vocabulary coverage (via parallel sentences) in-context is the most efficient method for improving LLM-based XLR translation. These linear regressions show that translation performance can be directly modelled by test set vocabulary coverage, and that the book's grammar explanations provide no *significant* advantage over its parallel sentences. See Appendix F for full statistics on our prompts' test set type coverage.

We then explore whether the improved translation scores can be attributed to a longer (or shorter) prompt, by testing for a relationship between prompts' total *tokens* and translation quality in terms of CHRF++ for BOOK$_{\{\text{all}/p/\neg p\}}$ with `Gemini`. The resulting linear models are not significant in either direction ($p = 0.997$, $p = 0.78$ into and from `kgv`, F-test), with no significant Pearson correlations.

---

[4]For details of these and following statistical tests, see Appendix B.

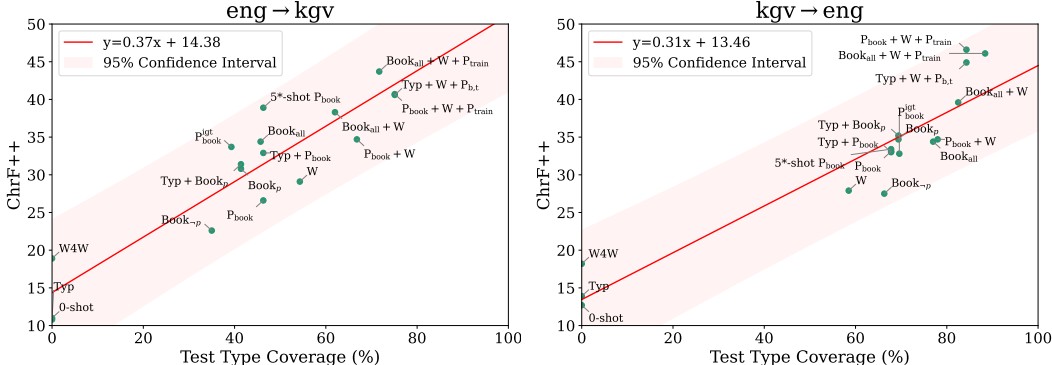

Figure 2: Regression models of CHRF++ score against test type coverage for `eng–kgv` and `kgv–eng` translation with `Gemini`. Prompt settings are labelled with abbreviations for clarity. The plots show that translation performance can be statistically modelled by test set vocabulary coverage.

The grammar book is therefore both a token-inefficient way to learn (with similar performance despite nearly 5x more tokens than `kgv` BOOK$_p$), and a cost-inefficient dataset to generate, compared to using its parallel sentences. The needle-in-a-haystack problem could partially explain this: with increasing context, retrieval of relevant information (i.e. similar parallel examples) becomes harder (Hsieh et al., 2024), so while BOOK$_p$ is a subset of BOOK$_{all}$, there is a greater ratio of relevant to irrelevant information in the prompt—assuming grammatical explanations cannot be effectively exploited for translation.

**Discussion** We note that our results do not indicate LLMs cannot understand books in general; rather, we find no quantitative evidence that the results here and in MTOB show LLMs can effectively exploit grammar books (or linguistic knowledge) *for translation*. Indeed, we show that LLMs can exploit grammatical information in the form of typology for more relevant, linguistically-focused tasks. More broadly, from an educational perspective, translation is a problem-solving task aiming to reach a goal state (translation) via a series of actions given an initial state (source) and optionally rules on applying actions. Humans tend to learn this kind of task more efficiently via worked-examples (van Gog et al., 2019), i.e. with explicit explanations, rather than pure discovery learning, meaning without explicit guidance (Mayer, 2004). Our results however indicate that for translation, LLMs learn more effectively from unannotated parallel examples (i.e. discovery) than from grammar principles with explained examples (i.e. example-based). Our results thus tentatively support a divergence between learning strategies for translation between human learners and LLMs learning in-context. We suggest that this may partially stem from prompts with parallel data aligning more closely with LLMs' instruction-tuning data than grammar book explanations.

# 6 CONCLUSION

We find no evidence that LLMs can effectively exploit grammatical explanations for low and extremely low-resource MT in Kalamang, Nepali, and Guarani, instead finding that LLMs rely on the parallel sentences within the book. This runs counter to the claim of prior work including MTOB which use grammar books to enable LLMs' performance on XLR tasks. We show that fine-tuning small MT models matches the performance of costly long-context LLMs. Further, we show statistically that grammatical explanations add no significant advantage above the increased type coverage they provide, and that grammar books are less token-efficient for prompting than parallel sentences. However, LLMs *can* exploit grammatical information, given an appropriate task—e.g. grammaticality judgment or IGT prediction—and more useful grammatical data in the form of our typological prompt, which achieves leading results on these linguistic tasks. We therefore emphasise the importance of task-appropriate data: parallel data for MT, and grammatical, preferably typological, knowledge for linguistic tasks. Moreover, we suggest data collection efforts for multilingual XLR tasks, at least for MT, are better focused on parallel data over linguistic description, which enables less costly, more token-efficient translation.

ETHICS STATEMENT

We emphasise that this work does not aim to address social problems, and instead investigates the empirical utility of grammar books as resources for XLR NLP. We operate on the assumption of continued consent of the Kalamang community to use their language in our research, as discussed in Tanzer et al. (2024). In Sections 2 and 3.5 we discuss the utility of our work for both linguists and L1 speakers, specifically relating to the IGT prediction task in its capacity to improve language documentation processes.

ACKNOWLEDGEMENTS

This work was funded in part by the UvA's *Language Sciences for Social Good* project, the City of Amsterdam, and the Netherlands Organization for Scientific Research (NWO) under project numbers VI.C.192.080 and 2023.017. The authors would like to thank members of the Language Technology Lab for many constructive discussions, particularly Vlad Niculae for detailed feedback on our paper. The authors are grateful for the helpful feedback provided by the anonymous reviewers.

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

## A  Kalamang Grammar Book Extract

In Figure 3, we provide a brief extract from the Kalamang grammar book (Visser, 2022), where the first paragraph exemplifies $\text{BOOK}_{\text{non-para}}$, and examples 17 and 18 show the format of $\text{BOOK}_{\text{para}}$.

> Clitics are mainly inflectional, and include postpositions as well as aspect and mood markers. Among the derivational clitics are attributive *=ten*, which derives adjectives from verbs but is also attested in non-verbal predicates (§6.3.5), and causative *ma=* (§11.4.4). Cliticisation always occurs after affixation. A derived noun, for example, can carry a postposition. *Lenget* 'villager' from *leng* 'village' and agent nominaliser *-et* becomes *lenget=at* when it is the last constituent of the object NP, as in (17). *Amkeiret* 'birth parent' from *amkeit* 'to give birth' and agent nominaliser *-et* can be inflected with animate lative *=kongga*, as in (18).

> (17)  *ma  sontum  leng-et=at  merengguen*
>       3SG person  village-NMLZ=OBJ gather
>       'He gathered the village people.'                    [narr27_3:17]

> (18)  *don  wa  me  se  amkeit-et=kongga*
>       thing PROX TOP IAM give_birth-NMLZ=AN.LAT
>       'This thing comes from the birth parent.'            [conv20_38:53]

Figure 3: A brief passage from Visser (2022), showing the format of $\text{BOOK}_{\text{non-para}}$ (above) and $\text{BOOK}_{\text{para}}$ (examples 17 and 18) explaining a morphological feature of Kalamang.

## B  Statistical Tests

As discussed in Section 5.1, we fit linear regression models to CHRF++ score with test set type coverage as the independent variable for `Gemini` eng⇌kgv translation experiments. We find the models are significantly useful in both directions according to the F-test, ($p \ll 0.005$). For `eng-kgv`: $F(1, 15) = 79.3, R^2 = 0.84, p = 2.3 \times 10^{-7}$, and for `kgv-eng`: $F(1, 15) = 98.1, R^2 = 0.87, p = 5.7 \times 10^{-8}$. The Pearson correlations of these results are also significant, where for `eng-kgv`: $r = 0.92, p = 1.1 \times 10^{-7}$, and for `kgv-eng`: $r = 0.93, p = 2.8 \times 10^{-8}$.

Finally, in modelling CHRF++ with prompt *tokens* as the independent variable for $\text{BOOK}_{\{all/p/\neg p\}}$, we find the resulting linear models are not significant according to the F-test. For `eng-kgv`, $p = 0.997, F(1, 1) = 0.00, R^2 = 0.00$; and for `kgv-eng`, $p = 0.78, F(1, 1) = 0.13, R^2 = 0.11$. There is no correlation between the number of tokens and the observed CHRF++ score.

## C  Test set analysis

To illustrate the weakness of the `kgv` test set, we generate `eng-xxx` test sets in: Dutch (`nld`), German (`deu`), French (`fra`), and Spanish (`spa`) using Google Translate[5], and test `Gemini`'s performance on these sets to find the upper bound. Table 6 shows the test set is weak and a score below 50 CHRF++ falls far below the observed high-resource upper bound. The 100 example set also falls well below standard translation test sets in size, usually 500-1000 examples (Costa-jussà et al., 2024). We addressed the issues of simplicity and size by testing the `npi` and `gug` FLORES test sets.

## D  Typological Feature Prompt

We provide an extract of the `kgv-eng` typological feature summary constructed from Grambank (Skirgård et al., 2023b) in Table 7.

---

[5] https://cloud.google.com/translate

Table 6: CHRF++ and BLEU scores of `Gemini` zero-shot tests on the translated 100-example `kgv` test set, plus our best `kgv` results.

| Setting | BEST | | 0-SHOT | | | | | | | | | |
|---|---|---|---|---|---|---|---|---|---|---|---|---|
| Language | kgv | | kgv | | nld | | deu | | fra | | spa | |
| Direction | → | ← | → | ← | → | ← | → | ← | → | ← | → | ← |
| CHRF++ | 43.7 | 46.6 | 11.0 | 12.7 | 80.5 | 73.7 | 80.1 | 67.1 | 87.8 | 72.2 | 84.0 | 73.1 |
| BLEU | 12.2 | 22.5 | 0.0 | 0.0 | 61.1 | 53.6 | 63.4 | 44.4 | 80.0 | 51.6 | 74.2 | 53.6 |

Table 7: An extract of our typological feature prompt TYP constructed from Grambank data, specifying features and descriptions for both source (`kgv`) and target (`eng`) languages where available.

---

The following typological features describe the grammatical features of Kalamang and English including word order, verbal tense, nominal case, and other language universals. Each feature is assigned a value that indicates the extent to which the language tends to exhibit that feature.

**Feature ID: GB020** Are there definite or specific articles?
Kalamang Value: absent, Code 0
Kalamang is coded 0 for this feature, meaning the feature is absent.
This feature indicates Kalamang does not obligatorily encode the grammatical function of definite articles.
English Value: present, Code 1
English is coded 1 for this feature, meaning the feature is present.
This feature indicates English obligatorily encodes the grammatical function of definite articles.
—

Below is a short summary of the grammatical feature, an explanation of the process for assigning the feature's code, and examples of the feature from other languages including interlinear glossed text.
—

**Are there definite or specific articles?**
**Summary** An article is a marker that accompanies the noun and expresses notions such as (non-)specificity and (in)definiteness. Sometimes these notions of specificity and definiteness are summed up in the term 'identifiability'. The formal expression is irrelevant; articles can be free, bound, or marked by suprasegmental markers such as tone. Articles are different from demonstratives in that demonstratives occur in a paradigm of markers that have a clear spatial deictic function. As demonstratives can grammaticalize into definite or specific articles, they form a natural continuum, making it hard to define discrete categories, but to qualify as an article a marker should be used in some cases to express definiteness without also expressing a spatial deictic meaning.
**Procedure** 1. Code 1 if there is a morpheme that can mark definiteness or specificity without also conveying a spatial deictic meaning.
2. Code 0 if the source does not mention a definite article and you cannot find one in examples or texts in an otherwise comprehensive grammar.
3. Code ? if the grammar does not contain enough analysis to determine whether there is a definite article or not.
4. If you have coded 1 for GB020 and 0 for GB021 and GB022, please write a comment explaining the position of the definite or specific article.
This is the end of the summary for feature GB020: "Are there definite or specific articles?".
—

**Feature ID:** [...]

—

This is the end of the typological feature summary for Kalamang and English.

## E   PROMPT EXAMPLES

To further clarify the difference between prompt settings, we provide brief excerpts in Table 8.

Table 8: Excerpts from various prompt settings for `kgv-eng` translation. All prompts also include the text from the 0-SHOT setting.

| Setting | Example |
|---|---|
| 0-SHOT | Kalamang is a language spoken on the Karas Islands in West Papua. Translate the following sentence from Kalamang to English: [source]
Now write the translation. If you are not sure what the translation should be, then give your best guess.
Do not say that you do not speak Kalamang. Do not say you do not have enough information, you **must** make a guess. If your translation is wrong, that is fine, but you have to provide a translation.
Your translation **must** be on the first line of your response, with no other text before the translation. Only explain your reasoning after providing the translation.
It is crucial that you **only** give the translation on the first line of your response, otherwise you will fail. Now write the translation:
Kalamang: [source] English: |
| 5*-SHOT | To help with the translation, here is a translated sentence with words similar to [source word] in a list of translated Kalamang-English reference sentences:
Kalamang: [source].
English translation: [target]
To help with the translation, here is a translated sentence... |
| WORDLIST | To help with the translation, here is a Kalamang-English word list:
Kalamang: =a = English: focus marker
Kalamang: a = English: filler
Kalamang: a'a = English: yes
Kalamang: adat = English: tradition
Kalamang: ade = English: pejorative interjection
Kalamang: adi = English: interjection of pain |
| PARA$_\text{book}$ | To help with the translation, here are some example Kalamang-English parallel sentences:
Kalamang: Bal se sorat koraru.
English translation: The dog has bitten the fish.
Kalamang: Mu kiem.
English translation: They run.
Kalamang: Ma reitkon purapi anat kamatet.
English translation: He sent me one hundred and fifty thousand rupiah. |
| PARA$_\text{book}^\text{IGT}$ | To help with the translation, here are some example Kalamang-English parallel sentences:
Kalamang: bal se sor=at koraru = Interlinear gloss: dog IAM fish=OBJ bite = English translation: The dog has bitten the fish.
Kalamang: mu kiem = Interlinear gloss: 3PL run = English translation: They run.
Kalamang: ma reitkon purap-i an=at kamat=et = Interlinear gloss: 3SG hundred fifty-OBJQNT 1SG=OBJ send=IRR = English translation: He sent me one hundred and fifty thousand rupiah. |
| BOOK$_\text{para}$ | To help with the translation, here is the full text of a Kalamang-English grammar book:
—
bal se sor=at koraru
dog IAM fish=OBJ bite
'The dog has bitten the fish.'
mu kiem
3PL run
'They run.' |
| BOOK$_\text{non-para}$ | To help with the translation, here is the full text of a Kalamang-English grammar book:
—
This is a description of Kalamang (ISO 639-3 code kgv, glottocode kara1499), a Papuan language of the Greater West Bomberai family. It is spoken by around 130 people in East Indonesia. The majority of speakers live on the biggest of the Karas Islands, which lie just off the coast of the Bomberai Peninsula in West Papua province. The language is known as Karas in older literature ... |

## F  PROMPT VOCABULARY STATISTICS

In Table 9, we show test set out-of-vocabulary (OOV) type counts (i.e. unique words) and corresponding test set type coverage in the input prompt for each setting. If the prompt includes a word that is in the test set in the target language, we count that as an in-vocabulary type, and words which do not appear in the prompt as OOV; our denotation of OOV is therefore unrelated to the model's vocabulary. We additionally include token counts (individual occurrences of types) for each prompt.

Table 9: Test set OOV type counts and type coverage, plus token counts, for all prompt settings in eng⇌kgv translation.

| | eng–kgv | | kgv–eng | | |
|---|---|---|---|---|---|
| Setting$_\downarrow$ | OOV | Coverage (%) | OOV | Coverage (%) | Prompt Tokens |
| 0-SHOT | 374 | 0.0 | 395 | 0.0 | 0 |
| W4W | 374 | 0.0 | 395 | 0.0 | 0 |
| WORDLIST (W) | 171 | 54.3 | 164 | 58.5 | 9011 |
| 5*-SHOT PARA$_{book}$ | 201 | 46.3 | 127 | 67.8 | 852 |
| PARA$_{book}$ | 201 | 46.3 | 127 | 67.8 | 15561 |
| + W | 124 | 66.8 | 87 | 78.0 | 24572 |
| + PARA$_{train}$ | 93 | 75.1 | 62 | 84.3 | 29407 |
| PARA$_{book}^{IGT}$ | 227 | 39.3 | 120 | 69.6 | 22686 |
| BOOK$_{all}$ | 203 | 45.7 | 91 | 77.0 | 99579 |
| + W | 142 | 62.0 | 69 | 82.5 | 108590 |
| + PARA$_{train}$ | 106 | 71.7 | 46 | 88.4 | 113425 |
| BOOK$_{para}$ | 219 | 41.4 | 121 | 69.4 | 18309 |
| BOOK$_{non-para}$ | 243 | 35.0 | 133 | 66.3 | 81270 |
| TYP 0-SHOT | 374 | 0.0 | 395 | 0.0 | 68426 |
| + BOOK$_{para}$ | 219 | 41.4 | 121 | 69.4 | 86735 |
| + PARA$_{book}$ | 201 | 46.3 | 127 | 67.8 | 83987 |
| + W + PARA$_{book+train}$ | 93 | 75.1 | 62 | 84.3 | 100581 |

## G  ADDITIONAL FINE-TUNING RESULTS

Table 10 shows translation results for fine-tuning the instruction-tuned Gemini on PARA$_{book}$ data, and tested in a 0-SHOT setting. Included are results with Llama-ft and fine-tuned NLLB. Fine-tuning the small MT model is more effective than tuning an LLM in this particular 0-SHOT setting.

Table 10: Translation results for eng⇌kgv with Gemini, Llama base, and NLLB, fine-tuned on the preprocessed PARA$_{book}$ data. We observe tuning a translation model is more effective than tuning an LLM (whether pretrained or already instruction-tuned) in this setting.

| | CHRF++ | | | | | |
|---|---|---|---|---|---|---|
| Setting$_\downarrow$ | | eng–kgv | | | kgv–eng | |
| Model$_\rightarrow$ | Gemini-ft | Llama-ft | NLLB | Gemini-ft | Llama-ft | NLLB |
| FT-PARA$_{book}$ | 20.2 | 18.5 | 34.2 | 19.3 | 23.0 | 28.6 |

## H  LIMITATIONS

In addition to those noted in the main paper, we acknowledge the following limitations of this work. While we combine the Kalamang test sets to give a 100 example set, this is still far below a standard test set for MT, often 1-2k sentences. In Kalamang, we are limited by the availability of additional data. However we do test Nepali and Guarani with the FLORES devtest set of 1012 examples which

provides more realistic low-resource translation settings. Nepali and Guarani experiments also address generalisation issues of focusing only on one XLR language. Regarding evaluation, we note that many differences in CHRF++ score were fairly small, and as reported in Kocmi et al. (2024) a difference in CHRF (note, not CHRF++) of 3.05 is required for more than 90% of humans to agree that a system is better than another in practice; this emphasises the need for future experiments and qualitative analyses (see Appendix I for a small scale qualitative analysis).

Further, the majority of our translation experiments are run on `Gemini-1.5-Flash`, an API-only LLM. Given the nature of our long-context experiments, we are necessarily limited in our choice of model—at the time of running experiments and to our knowledge, no other model family can handle context lengths over 200k tokens which is necessary for the entire Kalamang book. We run selected short-context experiments with the open-weight `Llama-3.1-8B` model to improve the generalisation of our results, and we leave tests with other long-context models to future work. We finally note that while ideally we would have a larger `kgv` test set, running long-context inference of paid API models for >1k examples becomes prohibitively expensive. This limitation applies to the entire method of long-context LLM prompting, justifying the fine-tuning of smaller, open-weight, local models for XLR translation instead—especially for members of these language communities who are unlikely to have access to large API models but may have access to free GPUs through services such as Google Colab[6] and Kaggle[7].

## I   QUALITATIVE EVALUATION

Table 11 shows 7 test set examples of Kalamang to English translation with various `Gemini` prompting settings. We note again that a qualitative evaluation of English to Kalamang translation is not possible without a Kalamang speaker among the authors. We also note that the test set examples have been available online from Dictionaria[8] (Visser, 2020) and its related Github repository[9] since November 2020. We argue this does not compromise our results, since we always compare performance with the book to 0-SHOT settings; whether or not the model has already seen the test set is less relevant if the 0-SHOT performance is extremely poor, as is the case for `kgv`. Let us now qualitatively discuss each one in turn.

In Example 1, 0-SHOT only translates the borrowed word 'fiber' and the name (visible to due capitalisation), but is otherwise irrelevant. 5*-SHOT gets some vocabulary correct such as 'boat' and 'grandfather', but misses the overall meaning. While BOOK$_{\neg p}$ manages some correct lexical translation, many words are incorrect and the overall meaning is lost. Both BOOK$_{\text{all}}$ and BOOK$_p$ get the general meaning correct, but BOOK$_p$ is more accurate, correctly generating 'two' and 'are' over 'is', and more naturally predicting 'the red one'. BOOK$_p$ is therefore marginally more grammatically correct and fluent, in relation to the reference target.

Example 2 shows predictably poor performance in the 0-SHOT setting. For 5*-SHOT, the model manages some correct lexical translations but the sentence-level meaning is lost. BOOK$_{\text{all}}$ and BOOK$_p$ get most of the meaning; however they both miss some lexical translation (e.g. 'sacrifice' rather than 'medicine') and incorrectly predict verb tenses. BOOK$_{\neg p}$ only correctly translates a few words (including 'child' and 'born'), and generates an unrelated sentence.

In Example 3, 0-SHOT is again an inadequate translation (despite being fluent). Here, the 5*-SHOT setting is also off-target in meaning, being unable to find a translation for 'Desili' and instead using it as a name. BOOK$_{\neg p}$ also fails to translate this word, and the output is irrelevant. BOOK$_{\text{all}}$ and BOOK$_p$ predict a similar meaning, close to the target; however, BOOK$_p$ correctly generates the tenses of past continuous 'planing' and the present simple 'cut', instead of the simple past 'planed' and 'went to cut'. Therefore here the parallel, glossed examples in BOOK$_p$ help to predict correct grammar moreso than the grammatical explanations in BOOK$_{\text{all}}$ and BOOK$_{\neg p}$.

Example 4 shows a largely irrelevant 0-SHOT translation, with the correct proper noun. 5*-SHOT gets the possessive 'father', and the meaning of 'one hundred', but the overall meaning is lost. The BOOK settings are similar and get different aspects of lexical and sentence-level meaning correct. BOOK$_{\text{all}}$ is

---

[6] https://colab.research.google.com/
[7] https://www.kaggle.com/code
[8] https://dictionaria.clld.org/contributions/kalamang#texamples
[9] https://github.com/dictionaria/kalamang/tree/v1.0

the worst among them, predicting an inadequate output. BOOK$_{\neg p}$ correctly translates the meaning of 'one hundred', but fails to translate 'walorkawat'; and BOOK$_p$ is the only setting to mostly correctly translate 'coconut leaves', but misses the meaning of 'father's family' and 'one hundred'.

In Example 5, 0-SHOT is completely wrong. 5*-SHOT and BOOK$_p$ outputs are identical, and close to the meaning but lack the reference's specificity. BOOK$_{all}$ and BOOK$_{\neg p}$ however both predict negation, and output a grammar book-style sentence showing the indeterminate gender of the pronoun with 'He/She', which is penalised against the reference; BOOK$_{\neg p}$ also misses the meaning of sickness.

Example 6 again illustrates the largely inadequate 0-SHOT performance. Here, the 5*-SHOT setting is fairly lexically accurate with 'beach' and 'tall' (against 'long'), but misses the sentence-level meaning. BOOK$_{\neg p}$ fails to translate 'beach' but gets some of the meaning; while BOOK$_{all}$ and BOOK$_p$ both get some aspects correct: the former keeps the beach's name but misses the word 'beach', and the latter misses the name but predicts 'beach'.

Finally, in Example 7 we see another failure of the 0-SHOT setting. With 5*-SHOT, the model gets the verbs correct but misses some vocabulary (i.e. 'the bay') and the general meaning. Both BOOK$_{all}$ and BOOK$_{\neg p}$ predict 'District Officer' for 'Camat', which is the Indonesian translation, while the reference denotes this as a given name, showing the model relying on previously observed but unrelated vocabulary when lacking a translation in the prompt. BOOK$_p$ is closer to the reference meaning for the first clause, though with the present perfect 'has come' instead of the simple past 'came', and misses some meaning in the second clause. BOOK$_{all}$ is further away from the reference in the second clause referring to 'what' rather than 'Camat/him', and BOOK$_{\neg p}$ misses 'the bay' and has an incorrect subject for 'know'.

In summary, 0-SHOT is predictably irrelevant but fluent; 5*-SHOT tends to give correct lexical translations with incorrect sentence-level meaning; BOOK$_{\neg p}$ predicts some higher-level meaning but lacks lexical translation adequacy; and BOOK$_{all}$ and BOOK$_p$ produce the best translations which are largely complete at the sentence-level, with BOOK$_p$ sometimes generating more precise grammar and lexical translations.

Table 11: Examples of source, target, and predicted outputs for 0-SHOT, 5*-SHOT, BOOK$_{\{all/p/\neg p\}}$ settings, in `kgv–eng` translation with `Gemini`.

| Setting | Output | |
|---|---|---|
| `kgv` src | Fiber eira metko, kon tama Dian taraunkin? Kon kerkapten. | |
| `eng` tgt | There are two fibre boats there, which one is Dian's grandfather's? The red one. | |
| 0-SHOT | The fiber is wet, is it Dian's fault? It's broken. | |
| 5*-SHOT | Is there a fibre boat there, where is Dian's grandfather? It's getting dark. | (1) |
| BOOK$_{all}$ | The fiber boat is there, which one is Dian's grandfather's? The one that is red. | |
| BOOK$_{p}$ | Those two fiber boats are over there, which one is Dian's grandfather's? The red one. | |
| BOOK$_{\neg p}$ | Is there fiber over there, which one is Dian's? It's the one with the cover. | |
| `kgv` src | Wa me kariak kaia kon untuk emumur kalo tumun amkeiret mu wara nanet. | |
| `eng` tgt | This is blood medicine for mothers, when they have given birth they drink this. | |
| 0-SHOT | You should be careful when you go out at night. | |
| 5*-SHOT | This one is a kind of food for women if a child is born they eat it. | (2) |
| BOOK$_{all}$ | This is the blood of a woman who has given birth to a child. | |
| BOOK$_{p}$ | This is a blood sacrifice for the women so that the children will be born. | |
| BOOK$_{\neg p}$ | This is a story about a child who was born on the island. | |
| `kgv` src | Desili koyet an koi panggutki tenaunat tawara. | |
| `eng` tgt | After planing, I use the panggut to chop the keel. | |
| 0-SHOT | The young man is very strong. | |
| 5*-SHOT | Desili is diligent and she is planting cassava. | (3) |
| BOOK$_{all}$ | After I planed, I went to cut the keel. | |
| BOOK$_{p}$ | After planing, I again cut the keel. | |
| BOOK$_{\neg p}$ | Desi is already here and she is asking for a drink. | |
| `kgv` src | Kalamang esun mu walorkawat reitkon. | |
| `eng` tgt | Kalamang's father's family have one hundred coconut leaf stems. | |
| 0-SHOT | The Kalamang people are very strong. | |
| 5*-SHOT | Kalamang's father is waiting for the hundredth day. | (4) |
| BOOK$_{all}$ | The Kalamang people are fishing for lobsters. | |
| BOOK$_{p}$ | The Kalamang man has many coconut leaves. | |
| BOOK$_{\neg p}$ | The Kalamang people have one hundred walorkawat. | |
| `kgv` src | Ma kaleuna ning. | |
| `eng` tgt | He has a kidney-disease. | |
| 0-SHOT | You are very kind. | |
| 5*-SHOT | He is sick. | (5) |
| BOOK$_{all}$ | He/She is not sick. | |
| BOOK$_{p}$ | He is sick. | |
| BOOK$_{\neg p}$ | He/She is not going. | |
| `kgv` src | Os Kamburkadok me kahen. | |
| `eng` tgt | The beach at Kambur is long. | |
| 0-SHOT | The Kamburkadok are coming. | |
| 5*-SHOT | The one on the beach is tall. | (6) |
| BOOK$_{all}$ | The sand on Kamburkadok is far away. | |
| BOOK$_{p}$ | The sand on the beach is far away. | |
| BOOK$_{\neg p}$ | The Kamburkadok is far away. | |
| `kgv` src | Camat mu lukta, in arep neko komahal. | |
| `eng` tgt | Camat and family came, we in the bay didn't know. | |
| 0-SHOT | The chief is sick, and the people are worried. | |
| 5*-SHOT | They came first, we don't know where they went. | (7) |
| BOOK$_{all}$ | The District Officer came to us, we don't know what's inside the bay. | |
| BOOK$_{p}$ | The Camat has come, we don't know where he is in the bay. | |
| BOOK$_{\neg p}$ | The district officer came here, but he doesn't know where we are. | |

