# OpenReview forum: "Can LLMs Really Learn to Translate a Low-Resource Language from One Grammar Book?"
_ICLR.cc/2025/Conference — ICLR 2025 Spotlight_

### Official Review · Reviewer_DQQ1 · 2024-10-30

**Soundness:** 4
**Presentation:** 4
**Contribution:** 2
**Rating:** 6
**Confidence:** 4

**Summary:**

This paper examines the contribution of a grammar book to building a machine translation system for an extremely low-resource language, Kalamang, that LLMs cannot have seen in training. In Kalamang, there is also parallel sentences and a dictionary. The authors evaluate LLMs by placing different mixtures of these language resources into a long context and evaluate the model's ability for in-context learning with the goal of translating test samples to and from English. They find that the additional grammatical explanations contribute little, if anything, on top of the parallel sentences, which themselves help the LLM learn to translate Kalamang. They similarly find that for Nepali, a much higher-resource language (but still low) where models have likely seen some text for, parallel sentences contribute to improving MT but not the grammar book. They also show this when finetuning NLLB, a translation-specific model, and show that grammar is also token inefficient in ICL in comparison to parallel sentences.

**Strengths:**

1. The paper is very well organized and is very clear to follow
2. The paper properly investigates a research question, with few, if any, holes in their experimental design as far as I can find
3. The presentation of the experiments and results are clear and the findings align logically with the numerical results
4. The authors incorporate novel methods (as well as new datasets) to understand the importance of grammatical explanations for low-resource MT

**Weaknesses:**

1. In general, the biggest weakness of the paper is the significance of its findings:

a. It is already established that parallel data, the most directly related samples to the MT task, are the most effective to training MT models. In many languages, parallel data is in fact the only form of labeled data that exists. For Nepali and Guarani, for example, it feels obvious that a grammar book would not help given that Gemini and Llama have likely implicitly seen many Nepali and Guarani texts in pretraining (or other highly related Indo-Aryan languages with similar grammar).

b. I understand that, by nature of targeting extremely low-resource languages, there are very few available data sources for such an experiment. But generalizing these results past Kalamang seems questionable, especially given that there is a singular grammar book, presumably written by a very small number of people. If possible, this paper would be strengthened by a few more languages (more similar in task than Nepali). I would also suggest more context on grammar books themselves be provided and how frequent these are for extremely low-resource languages.

c. Title is a bit misleading in that it asks whether using grammar books are useful, all to prove that not really. The title moreso implies that the paper would show that grammar books are useful.

2. In comparison to ACL venues, ICLR attendees will typically have less linguistics background. I am concerned the discussion of more specific linguistics topics (e.g. interlinear gloss text, grammar & typology) might be not well suited for this audience. I will leave that to the AC to discern, but also advise authors modify the Related Work & Methodology section to take that into account.

**Questions:**

1. Line 44: how does the quantity of data available make Kalamang not unique ? I would better explain why the findings on Kalamang are relevant to many other languages
2. I would discuss the fact that you are using in-context learning (ICL) more clearly in the intro or even abstract
3. I would add more references to explain/justify the methodology in 3.3 and 3.5
4. Line 215: parallel data annotation is typically very rigorous; annotators have to follow specific instructions or for scraping, sentence alignment has to be in a particular way or else the sample is thrown out. How clean & consistent are the parallel samples that come from your heuristic filter ? How did you determine what was appropriately parallel examples vs an explanation when no Kalamang speakers were involved in the project ? I would add more details about this heuristic filtering process.
5. Notation for tables is confusing and takes a while to digest. It might be worth including quick natural language names like instead of Book_p, putting "Grammar Book, parallel subset"
6. Table2: how is Gemini able to do 11&13 ChrF when it has never seen Kalamang before ?

---

> ### Author Response · Authors · 2024-11-20
>
> We thank the anonymous reviewer for their valuable feedback. We respond to each point in turn below.
>
> **Weakness 1a**: The standard setting for training seq2seq MT models from scratch indeed works with parallel data; however our main experimental setting is distinct from that standard scenario, since for Kalamang we have a grammar book and extremely limited parallel sentences, and we use LLMs in-context to leverage this non-standard data. We emphasise that the purpose of this paper is to revisit claims in previous work, primarily Tanzer et al. (ICLR, 2024) who claim that a single grammar book in-context may suffice for LLMs to learn to translate an (unseen) language. Our findings call these claims into question: we show that LLMs only really learn from the parallel examples within the book. We show this via a thorough analysis of different prompting setups, and further we introduce a novel approach that effectively uses typological knowledge instead of grammar books for grammatical tasks. Our tests on Nepali (and Guarani) show that these findings hold in both XLR and low-resource translation scenarios – LLMs fail to exploit grammatical explanations in general, rather than failing to use the explanations because the language is already seen. Our findings are therefore a contribution on the question of grammar books for LLM-based translation, an area which is receiving increasing research interest (Hus & Anastasopoulos. 2024, Tanzer et al. 2024, Zhang et al. 2024, Ramos et al. 2024).
>
> **Weakness 1bi**: Regarding generalisation to other low-resource languages, we add experiments on Guarani with a different grammar book (Estegarribia 2020, from UCL Press’s series “Grammars of World and Minority Languages”) to improve our findings’ generalisability. Guarani is similar in task to Nepali, given the similar availability of parallel data, plus a high quality test set (FLORES). Initial results in Guarani show the same pattern of parallel sentences providing most or all useful signal, and few-shot sentences are again the most effective prompt. (See the response to review 5eBi for initial results tables). We will add these results to our updated paper.
>
> **Weakness 1bii**: It is not entirely clear to us what is meant with “a form that LLMs are not expecting”, however, we use a standard form of grammar text: the Kalamang grammar is from MIT Language Sciences Press which has strict formatting and content guidelines for linguistic texts, and thus we expect this book is representative of grammar books in general. Second, while we cannot comment on what LLMs are expecting, we hypothesise that instruction-tuned models would perform better on prompts that are more similar to their instruction-tuning dataset. Given our discussion on p.10 (where we suggest LLMs learn more effectively from unannotated parallel examples than from grammar principles/explanations), we might suggest these LLMs were instruction-tuned on data including few-shot parallel examples, and not with grammar book-like explanations; whether or not this is the case, our finding holds that current LLMs are unable to exploit the grammatical explanations in grammar books and rely mainly on the parallel sentences. We will highlight this point in our Discussion.
>
> **Weakness 1c**: Regarding our title, “Can LLMs Really Learn to Translate a Low-Resource Language from One Grammar Book?”, the fact that it includes “really” within a question standardly implies that the speaker doubts the question’s premise. The premise here is the claim by prior work including Tanzer et al. (2024) that prompting LLMs in-context with grammar books enables the model to translate from and into an unseen, XLR language. Our paper revisits these claims, and our title implies that our results doubt these claims, instead showing grammar books do not really help LLMs to translate in this case (but their parallel sentences do).
>
> **Weakness 2**: We accept that the linguistics-specific topics in this paper may require some more introduction to some readers; to this end, we will further clarify glossing and typology in the Related Work and Methodology in our updated paper, with sufficient information to understand that a) IGT prediction is a well-motivated, grammar-focused task, and b) typological information is a useful and language-independent form of linguistic knowledge. We also provide an excerpt of our typological prompt in Appendix B, Table 6, for the interested reader. Additionally, we note in-depth linguistic knowledge is not required to understand the research questions, experiments, or main contributions of our work. Finally, one of our work’s central goals is to revisit claims made first in Tanzer et al. (2024), an ICLR 2024 spotlight paper; we therefore expect the audience to be appropriate.

---

> > ### Author Response · Authors · 2024-11-20
> >
> > [Continued due to character limit]
> >
> > **Question 1**: Regarding the applicability of our findings beyond Kalamang, we will clarify in the introduction that Kalamang’s resources are comparable to many other XLR languages, which, according to Joshi et al. (2020), make up a majority of the world’s languages.
> >
> > **Question 2**: On the discussion of in-context learning, we will clarify this in our updated paper’s introduction.
> >
> > **Question 3**: Regarding the justification of methodologies in 3.3 and 3.5, we will add the following references: Zhao et al. (2020), who introduced a standard setup for testing and evaluation of IGT prediction via morpheme accuracy; and McMillan-Major (2020) which similarly details a testing paradigm for IGT, including Stem and Gram F1 scores. The central reference is Ginn et al. (2023), which is a shared IGT prediction task and builds on the prior references; we base our own test set and evaluation paradigm on this work.
> > For typology in 3.5, we introduce a novel method for incorporating typological information in a textual format; to our knowledge there is no reference for prior methods here. However, we note the parallels with methods which use typological features for language embeddings (cf. Oncevay et al., 2020); the benefit of our setup is that it does not require any secondary model training for training embeddings. We will include details to this effect in our updated paper.
> >
> > **Question 4**: On the question of parallel data extraction: As shown in Appendix A, in the grammar book there is a clear distinction between parallel examples and everything else (i.e. explanations). Parallel examples are clearly aligned 1:1 (with glosses), and have clear formatting, numbers, and ID codes. We additionally filter word pairs present in tables and lists in the book based on formatting, but not those in running text. As discussed in our Related Work, we follow Nordhoff & Kramer (2022) in our parallel example extraction method, which relies on the text formatting (hence heuristic), and thus requires no manual labelling or knowledge of Kalamang. Our parallel examples are both clean and consistent because they are extracted from structured text. We will update our description of the filtering process in our updated paper.
> >
> > **Question 5**: We will update abbreviations in Table 2 for improved clarity, expanding $BOOK_{p}$ to $BOOK_{para}$, $BOOK_{\neg p}$ to $BOOK_{non-para}$, $P_{book}$ to $PARA_{book}$, and $P_{book}^{IGT}$ to $PARA_{book}^{IGT}$ respectively. The capitalised text indicates the form of the data (i.e. grammar book text, or processed parallel data with or without glosses) and the subscript indicates the subset of this data (e.g. $PARA_{train}$ = processed parallel data, from the training set, not the book). We will add this explanation in our updated paper in 4.1, and hope this explanation is clearer.
> >
> > **Question 6**: Regarding ChrF++ scores of zero-shot tests: an arbitrary output (even if produced in English for an eng–kgv test sentence with a kgv reference) often achieves a ChrF++ score of 10-15 due to random character overlap, and does not indicate any knowledge of the target language. Additionally, in some cases the models copy capitalised words (which are likely to be proper nouns) to the output. (NB The lower scores for Llama are due to the model occasionally producing a null output, in the more difficult eng–kgv direction.)
> >
> > We hope these comments and amendments address your concerns and look forward to your response. We will upload the updated paper shortly.

---

> > > ### Comment · Reviewer_DQQ1 · 2024-11-24
> > > **Response to Author Rebuttal**
> > >
> > > I find the rebuttal from the authors to be very reasonable, thank you for the thorough response. My few concerns of their experimental design were addressed and while I am still not fully convinced by the generalizability of this method, the additional context provided certainly supported their case. The addition of Guarani, however, is not too helpful as it resembles moreso Nepali in terms of resource-level in comparison to the extremely low-resource Kalamang.
> > >
> > > The contribution, however, still seems very narrow to me, especially when it comes to relating to ICLR audience.
> > > I will amend my review.

---

### Official Review · Reviewer_5eBi · 2024-11-02

**Soundness:** 4
**Presentation:** 4
**Contribution:** 3
**Rating:** 8
**Confidence:** 4

**Summary:**

This paper evaluated whether LLMs can translate extremely low-resource languages using a grammar book. The findings reveal that parallel sentence examples are crucial for translation, while grammatical explanations offer little benefit. Tests on English-Kalamang and Nepali confirm that translation quality improves with parallel data rather than grammatical descriptions. For linguistic tasks like grammaticality judgment, a typological feature prompt is more helpful. The authors recommend focusing on parallel data collection over grammar descriptions for effective translation in low-resource languages.

**Strengths:**

1.  The paper showed a novel study that whether grammar books alone support translation for extremely low-resource (XLR) languages or not. By isolating grammar book components (e.g. parallel sentences vs. grammatical explanations), they illustrated a comprehensive evaluation, complemented by introducing typological prompts for linguistic tasks, a creative addition for XLR NLP.

2. By the controlled experiments across different closed-source (Gemini), open-source LLMs (Llama 3.2 8b), and NLLB NMT model, They found out that LLMs can't effectively learn from grammatical explanations for XLR MT in Kalamang and Nepali, relying instead on the parallel sentences. Also, We showed that fine-tuning NLLB model matches the performance of LLMs with long context. Additionally, LLMs can utilize grammatical information effectively when applied to suitable tasks, e.g. grammaticality judgment or IGT prediction, especially when enhanced with targeted grammatical data like our typological prompt, which achieves top performance on these linguistic tasks.

3. The paper is well-written, coherent with comprehensive analysis.

**Weaknesses:**

1. While the paper effectively examines the Kalamang and Nepali languages, its findings may not generalize across the diversity of low-resource languages. XLR languages can vary significantly in structure, script, and available resources, and focusing only on these two could limit the applicability of the conclusions. It would be nice to do similar analysis on low-resource languages too.

2. The paper introduces typological prompts as a promising way to improve linguistic tasks but does not compare this approach against other techniques that incorporate linguistic typology (e.g., embeddings based on typological features from databases like [WALS](https://arxiv.org/abs/2010.03920))

**Questions:**

Please refer to the weaknesses.

---

> ### Author Response · Authors · 2024-11-20
>
> We thank the anonymous reviewer for their useful feedback, and respond to the comments below.
>
> **Weakness 1**: Regarding the generalisation of our results to other low-resource languages, we add experiments on Guarani ($\texttt{gug}$) which has a grammar book available of a similar length to Kalamang, and a FLORES test set. Initial results on filtered $BOOK_{p}$ and $BOOK_{\neg p}$ subsets below confirm what we observed for Nepali: most useful signal stems from the parallel data, with the grammatical explanations showing little added utility, and all book settings showing little to no improvement over 0-shot tests. This holds for both Gemini and Llama. Given this, we would expect similar results for other low-resource and XLR languages. Further, while Kalamang is an XLR language, Nepali is more accurately classified as a low-resource language which has some level of parallel data available; and Guarani also falls in this category. This difference is visible in the zero-shot performance of our LLMs (primarily Gemini) on Kalamang against Nepali and Guarani. Testing on Guarani also controls for script since like English it uses Latin script (with additional diacritics), whereas Nepali uses Devanagari script. We will add these results to our updated paper.
>
> ---
> **English--Guarani**
>
> | Setting          | Gemini-1.5-Flash | Llama-3.1-8B |
> | ---------------- | ---------------- | ------------ |
> |                  | ChrF++           | ChrF++       |
> | 0-shot           | 26.6             | 6.1          |
> | 5*-shot          | 29.2             | 13.7         |
> | $BOOK_{all}$     | 22.2             | --           |
> | $BOOK_{p}$       | 25.8             | 6.7          |
> | $BOOK_{\neg p}$  | 19.3             | 5.6          |
> | Typ 0-shot       | 21.1             | 4.3          |
> | Typ + $BOOK_{p}$ | 21.9             | --           |
>
> ---
> **Guarani--English**
>
> | Setting          | Gemini-1.5-Flash | Llama-3.1-8B |
> | ---------------- | ---------------- | ------------ |
> |                  | ChrF++           | ChrF++       |
> | 0-shot           | 41.3             | 23.6         |
> | 5*-shot          | 43.1             | 23.4         |
> | $BOOK_{all}$     | 38.7             | --           |
> | $BOOK_{p}$       | 41.8             | 11.8         |
> | $BOOK_{\neg p}$  | 34.5             | 10.1         |
> | Typ 0-shot       | 33.9             | 23.4         |
> | Typ + $BOOK_{p}$ | 34.5             | --           |
>
> ---
> **Weakness 2**: While using typological (or language) embeddings seems like a reasonable thing to try, they have been shown to have limited success (Ponti et al. 2019, Ustun et al. 2022) in other NLP tasks. Second, our typological prompt is a drop-in replacement for $BOOK_{\neg p}$, and is used in combination with $BOOK_p$; it is not immediately clear how we would combine embeddings and the textual input $BOOK_p$. We also note that adding complexity to the experimental setup (e.g. using specialised models) runs counter to our goal of using already available, human-readable resources to rapidly adapt a general-purpose model to new language pairs – i.e. in-context learning with LLMs. This comparison with typology/language embeddings is oblique to our main research question, and is a more substantial experiment than running a baseline model, therefore we leave further investigation to future work.

---

> ### Author Response · Authors · 2024-11-28
>
> Dear Reviewer 5eBi,
>
> We would like to thank you once again for your helpful feedback, and we would appreciate it if you'd let us know if our response and revisions to our paper address your concerns.
>
> Thank you!

---

### Official Review · Reviewer_nVmi · 2024-11-08

**Soundness:** 4
**Presentation:** 4
**Contribution:** 4
**Rating:** 8
**Confidence:** 4

**Summary:**

The study presents a very thorough revision of the claim that prompting LLMs with grammar books of an unseen language enables the model to translate from/into that language. The main claim is that actually the translation examples in grammar books are the key support of handling the language, while grammatical explanations do not bring any improvement. For that purpose the work uses grammatical books for Kalamang and Nepali and presents contrastive experiments with several language models (Gemini, Llama 3.1, other more specific models), covering prompting, fine-tuning and LoRA. Besides splitting the grammar books into parallel examples and everything else the authors also include a list of words and other supplementary material. In addition to the task of translating, the grammatically judgement is also included, where grammatical descriptions are useful unlike with translation.

Although the study is narrowly targeted, it presents very clearly and exhaustively verified conclusions on the capability of LLMs, showing once again that their behavior is more akin to elaborate pattern matching, rather than comprehension and thinking.

**Strengths:**

* wide coverage of materials, models and approaches
* presented clearly (except for the complexity of results in Table 2) and convincingly

**Weaknesses:**

* minor weakness: there are so many different models and comparisons that Table 2 is hard to understand, the reader has to go back and forth between the table and section 4.1 to remind, what the abbreviations mean -- perhaps there is a way to improve the readability?

**Questions:**

* would you expect different conclusions if the prompted / tuned LMs were much smaller?, what about larger models?

---

> ### Author Response · Authors · 2024-11-20
>
> We thank the anonymous reviewer for their helpful feedback. Regarding your comments:
>
> **Minor weakness**: We will update abbreviations in Table 2 for improved clarity, expanding $BOOK_{p}$ to $BOOK_{para}$, $BOOK_{\neg p}$ to $BOOK_{non-para}$, $P_{book}$ to $PARA_{book}$, and $P_{book}^{IGT}$ to $PARA_{book}^{IGT}$ respectively. The capitalised text indicates the form of the data (i.e. grammar book text, or processed parallel data with or without glosses) and the subscript indicates the subset of this data (e.g. $PARA_{train}$ = processed parallel data, from the training set, not the book). We will add this explanation in our updated paper in 4.1.
>
> **Question**: Regarding expectations for different model sizes, we would expect similar conclusions that parallel data provides most or all useful translation signal. Further, we know Gemini-1.5-Flash is substantially larger than Llama-3.1-8B (given the smaller Gemini-1.5-Flash-8B model), thus our results suggest larger models show better handling of long-context for translation tasks. We expect models smaller than 8B would show worse performance than Llama-3.1-8B, while models towards 70B and larger may show better results than Gemini-1.5-Flash. Indeed, in the Gemini-1.5 Technical Report (https://arxiv.org/abs/2403.05530), the authors observe translation quality improvements for the larger 1.5-Pro model over 1.5-Flash, suggesting performance on this task, including handling of a long-context grammar book, scales with model size.

---

> ### Author Response · Authors · 2024-11-28
>
> Dear Reviewer nVmi,
>
> We would like to thank you once again for your helpful feedback, and we would appreciate it if you'd let us know if our response and revisions to our paper address your comments.
>
> Thank you!

---

### Author Response · Authors · 2024-11-21

We thank all reviewers for their helpful and thorough feedback. In particular, reviewer nVmi highlighted our wide coverage of materials, models and approaches; reviewer 5eBi noted the novelty of our methods and findings; and reviewer DQQ1 emphasised the comprehensiveness of our experimental design and clear presentation of results.

We have revised our paper to address the comments and questions from the reviewers. In particular, regarding the generalisation of our results to other languages (noted by DQQ1 and 5eBi), we add experimental results and analysis for Guarani (Table 3), a seen low-resource language which is similar in task to Nepali and also has a high-quality FLORES test set. Our prior findings generalise to this new language for both Gemini and Llama models: current LLMs fail to exploit the grammatical explanations in grammar books for low-resource languages, and parallel data provides most of the useful signal for translation. Responding to reviewer DQQ1, we clarify the introduction of typology and interlinear glossed text and add appropriate references for our method; and finally we update the experimental abbreviations in Table 2 for clarity (noted by nVmi and DQQ1).

We hope these amendments address your concerns and look forward to your response.

---

### Meta-Review · Area_Chair_BzME · 2025-01-03

**Metareview:**

This is a nice paper that challenges recent claims about LLMs' ability to learn translation from grammar books alone, using as the use case Kalamang (extremely low-resource, XLR) and Nepali (low-resource but seen). Through experiments with different models (Gemini, Llama, and a fine-tuned NLLB MT system), the authors show that parallel sentence examples from the text  rather than the book’s grammatical explanations drive translation performance. The authors also check whether prompting the model with typological information about an XLR language can help in linguistic tasks, and find improvements inconsistent, casting more doubt on LLMs’ ability to effectively use grammatical information for machine translation.
Reviewer DQQ1 had some concerns regarding limited generalizability beyond the studied languages, particularly since only Kalamang is truly unseen XLR language. The authors address this in part by providing additional experimental results from Guarani. While reviewer DQQ1 maintains some reservations about the paper’s contribution, both reviewers nVmi and 5eBi strongly support acceptance.

Overall, despite the concerns about generalizability, I believe this work provides a useful data point w.r.t previous claims from Tanzer et al. (2024). The paper presents strong empirical evidence that parallel examples, not grammatical explanations, drive translation performance and it gives clear practical guidance on how to allocate resources for XLR language tasks. I recommend acceptance.

**Additional Comments On Reviewer Discussion:**

See above.

---

### Decision · Program_Chairs · 2025-01-22

Accept (Spotlight)